# The Role of Bond Functions in Describing Intermolecular Electron Correlation for Van der Waals Dimers: A Study of (CH_4_)_2_ and Ne_2_

**DOI:** 10.3390/ijms25031472

**Published:** 2024-01-25

**Authors:** Bogdan Rutskoy, Georgiy Ozerov, Dmitry Bezrukov

**Affiliations:** 1National Research Centre “Kurchatov Institute”, Moscow 123182, Russia; bogdan.rutskoy@itep.ru; 2Institute of Nuclear Physics and Technology, National Research Nuclear University “MEPhI” (Moscow Engineering Physics Institute), Moscow 115409, Russia; 3Chemistry Department, M.V. Lomonosov Moscow State University, Moscow 119991, Russia; dsbezrukov@gmail.com

**Keywords:** density matrix, dispersion function, van der Waals complexes, bond functions, electronic correlation

## Abstract

We present a study of the intermolecular interactions in van der Waals complexes of methane and neon dimers within the framework of the CCSD method. This approach was implemented and applied to calculate and examine the behavior of the contracted two-particle reduced density matrix (2-RDM). It was demonstrated that the region near the minimum of the two-particle density matrix correlation part, corresponding to the primary bulk of the Coulomb hole contribution, exerts a significant influence on the dispersion interaction energetics of the studied systems. As a result, the bond functions approach was applied to improve the convergence performance for the intermolecular correlation energy results with respect to the size of the atomic basis. For this, substantial acceleration was achieved by introducing an auxiliary basis of bond functions centered on the minima of the 2-RDM. For both methane and neon dimers, this general conclusion was confirmed with a series of CCSD calculations for the 2-RDM and the correlation energies.

## 1. Introduction

The weak interactions between closed-shell atoms and molecules are of fundamental importance for the accurate description of a great variety of systems in a range of fields. Important examples are the following: molecular biology and genetic engineering [1,2,3,4,5,6,7,8,9,10,11,12,13,14,15,16,17], in which stacking interactions, as well as the folding of protein and chromatin compounds, are highly dependent on hydrogen bonds and dispersion interactions. They also play an important role in the following: drug design and medicine [18,19,20], with the docking largely governed by the formation and breaking of non-covalent and non-ionic bonds; supramolecular chemistry, in which the spatial organization of constituent molecular systems is determined by weak covalent interactions; the chemistry of materials, in which the crystallographic structure is often determined by non-covalent interactions; catalytic chemistry, in which the adsorption of compounds and their distribution on a surface are strongly influenced by weak interactions; and molecular chemistry and chemical reactivity.

The accurate parameterization of intermolecular interactions in complex molecular systems, such as proteins [10], often relies on high-precision calculations involving small-model systems. Nevertheless, attaining an error level below 1% for intermolecular interactions represents substantial challenges. One of these pertains to the accurate incorporation of intermolecular electronic correlation contributions [21]. For instance, studies on potential energy surface (PES) cross sections for a methane dimer require the use of high-quality post-Hartree–Fock treatment, including the coupled clusters or second-order perturbation theory applying Pople and Dunning basis sets [22,23]. In this case, a basis set saturation (with up to 1% accuracy) has been achieved using the CCSD(T) method in the aug-cc-pVQZ basis set, yielding an interaction energy of 0.51 kcal/mol in the case of a complete basis set limit [23,24].

This highlights the importance of choosing a strategy to account for electron correlation effects that offer a balanced compromise between efficiency and accuracy for the given energy range. In particular, choosing methods to facilitate the slow-correlation energy convergence with respect to the basis set size [25,26] approaching the complete basis set (CBS) limit that mitigate the basis set superposition error (BSSE) [27,28,29] becomes essential. This issue and related issues involve the development of strategies to construct specially enhanced one-electron basis sets [21] designed to address the intermolecular correlation. Finally, yet another role in the quantitative description of these interactions is played by a finely tuned scheme employing the electron structure methods, which are usually represented using a mixture of perturbation theory, variational, or coupled clusters approaches.

The particular problem caused by the relatively small interaction energies concerns the convergence of the applied strategy in terms of the intrinsic approximation level of the quantum chemical strategy involved and the basis set sizes, in which the latter overlaps with the previously mentioned requirements of the special one-electron bases. An approved way of addressing the basis set saturation problem, in terms of the basis set size and the maximum angular momentum, is referred to as the bond functions approach [21,30]. With this approach, the additional dummy atomic center with the specially designed set of diffuse atomic functions is placed between molecular fragments to improve the description of the wave function nodal structure, as well as the overall behavior in this region. This goal partially overlaps with the focus of the explicitly correlated methods that are intended to describe the most pronounced form of the same effect in the whole space of the electronic coordinates (see, e.g., [31]). Illustrating the performance of this method, the basis set saturation with respect to the energy results is already attained at the cc-pVDZ level in the cases of the first- and second-row element compounds if the additional set of bond functions [33221] is invoked [23,32,33]. This effect has been most extensively studied in the context of a helium dimer, for which a comprehensive analysis of the correlation effects of the dispersion component of the two-particle wave function is available [21,32].

Although the bond function technique represents certain advantages, there is a high level of arbitrariness in the choice of the parameterization, including the dummy center position, as well as the auxiliary set size. Despite the fact that, for some noble gas diatomics, the bond functions centering issue was found to be of minor importance [34], the situation turned out to be more complex in the general case (e.g., see the profound benchmarking for the centering strategies in [35] against the A24 [36] and S66 [37] data sets). While existing heuristics to select the dummy center position as various averages of the monomer positions [38,39,40,41] generally lack theoretical rigor [35], regular analysis of auxiliary set parameterization [42] should take into account the effects of successive basis extension, optimizing the integral correlation characteristics under question with respect to the auxiliary function positions and exponents throughout the entire geometric parameter domain of interest, which becomes too computationally expensive. An alternative approach is to consider the optimization of basis parameters for the description of local correlation effects, rather than the integral energetic characteristics. For helium, this approach can be considered when analyzing interatomic interaction energy in the form of the expansion of the SAPT theory [43], which, in this case, includes the interatomic Hartree–Fock (HF) energy, as well as the intra-atomic and interatomic correlation contributions. In particular, the related contributions can be studied using the so-called dispersion function [44], which represents the part of the explicitly correlated genimal solution that is responsible for the monomers’ interaction. An alternative and more universal tool for the analysis of correlation effects is provided with the two-particle density matrix (2-RDM), the related cumulants, and the pair density functions [30,44,45,46]. It is well known that, for a nonrelativistic many-electron system that is non-interacting with an external magnetic field, the spin-averaged pair density contains the complete information about all the correlation effects [47]; this determines the quantity used to suggest the choice of the additional basis functions’ placement and “shape” parameters. Hence, the treatment of the auxiliary function parameters in the framework of 2-RDM analysis not only allowed us to bypass some of the expensive steps in interaction energy optimization [42] but also brought the pure two-electron correlation effects’ consideration and the problem of spatial compactness relevant to the standard Gaussian or Slater-type basis sets to common ground.

In the present work, we address these questions from a quantum chemical point of view, inspecting the electron correlation at the level of the two-electron density reduced spin averaged density and density matrix behavior. We show that placing the bond function of the domain in the coordinate space where pair density undergoes strong changes is a prerequisite for grasping the bond functions’ methodological advantages. To this end, we applied the response theory approach to construct effective the CCSD theory density matrices [48,49,50,51,52] for the methane and neon dimers in their ground states and the equilibrium geometries that were optimized at the same level of this theory. The resulting dependencies were applied to recover the spin-averaged two-electron density, which was, further, simply called pair density; the diagonal values of this density were addressed to adjust the choice of the basis set disposition.

In Section 2.1, we commence by outlining the relevant theoretical background, highlighting the role of the relaxed density matrices for correlation energy estimation in terms of the diagonal Hellmann–Feynman theorem for the non-degenerate level. In Section 2.2, we represent the main computational scheme for the calculation of the pair densities and other response properties, and in Section 2.3, we discuss and analyze our obtained data.

## 2. Results and Discussion

### 2.1. Theory

#### 2.1.1. Reduced Density Matrices in the CCSD Method

In the framework of the one-dimensional model space coupled cluster theory [53], the ground state solution of the stationary Schrödinger equation (H−E)|Ψ〉=0 within the given number of electrons *N* and abelian symmetry was searched using the following form:$$|\Psi \rangle =\Omega |\Phi \rangle =e^{T}|\Phi \rangle ,$$
where $|\Phi \rangle =\underset{|\Phi_{0}\rangle =|i_{1}\rangle \wedge \cdots \wedge |i_{N}\rangle }{\mathrm{argmin}}E_{\mathrm{HF}}\left(\Phi_{0}\right)$ is the Hartree–Fock physical vacuum, a wave operator Ω establishes correspondence the exact |Ψ〉 and model |Φ〉 vectors, and a cluster operator assumes the n-particle expansion $T=T_{1}+T_{2}+\ldots$, where the subscript enumerates the homogeneous components with respect to the excitation degree, which is restricted to 2 for the CCSD case. The one-particle unitary transformations entering the HF minimization have a generic form, $e^{\kappa }$, with an antihermitian one-electron operator κ and can be explicitly involved in the wave vector parameterization: $|\Psi \rangle =e^{-\kappa }e^{T}e^{\kappa }|\Phi \rangle =e^{-\kappa }e^{T}|\Phi \rangle$. Using the fact that $e^{T}$ is invertible allowed us to reformulate the problem in terms of the doubly dressed CC Hamiltonian ${\bar{H}}_{-\kappa ,T}=e^{-{\mathrm{ad}}_{T}}e^{{\mathrm{ad}}_{\kappa }}H=e^{-T}e^{\kappa }He^{-\kappa }e^{T}$ as
(1)$$\begin{array}{c} ({\bar{H}}_{-\kappa ,T}-E)|\Phi \rangle =0. \end{array}$$

Assuming the HF problem to be solved, and projecting Equation (Equation 1) on the space of excitations of the same degrees engaged in *T*, one can obtain the classical CC equation:(2)$$\begin{array}{c} \langle \Phi \left|\left(1+\Lambda^{+}\right)\bar{H}\right|\Phi \rangle =E_{\mathrm{CC}}, \end{array}$$
where $\bar{H}={\bar{H}}_{0,T}$, and Λ is an arbitrary linear combination of the excitation operators of the mentioned form. The first way, which leads to so-called unrelaxed densities, completely neglects the *S* part, which results in noticeable inaccuracies for the obtained densities and related quantities, although the resulting functional is still a state. It is a well-known result that the density matrices or states in the non-hermitian or hermitian formulations [45,54] can be approximated using the Lagrangian variational approach [55]. With this technique, the coupled clusters’ energy $E_{\mathrm{CC}}=\langle \Phi \left|\bar{H}\right|\Phi \rangle$ and the amplitude equations are to be summarized in the unique Lagrangian functional $L_{\mathrm{CC}}=\langle \Phi \left|\left(1+\Lambda^{+}\right){\bar{H}}_{-\kappa ,T}\right|\Phi \rangle$, where the operator $\Lambda^{+}$ represents the summation of the amplitude equations with the Lagrangian multipliers. Then, the stationarity conditions for $L_{\mathrm{CC}}$ with respect to the *T* amplitudes then provide the desired solution for the operator Λ. Additionally, the related equations represent the variational formulation of the CC theory treatment, which allowed us to apply the diagonal Hellman–Feynmann theorem for a non-degenerated case, providing a convenient method for evaluating the $E_{\mathrm{CC}}$ gradients.

Adapting the outlined consideration for the closed shell case of the CCSD approach, one should restrict the cluster operator to the spin-averaged one- and two-particle components:$$\begin{array}{c} T_{1}=\sum_{ai}t_{i}^{a}E_{a}^{i},\;T_{2}=\frac{1}{2}\sum_{abij}t_{ij}^{ab}E_{a}^{i}E_{b}^{j}, \end{array}$$
where $E_{ai}$ is a spin-averaged excitation operator from the *i*-th occupied orbital to the *a*-th virtual one. Next, the solution of the variational equations provides the complete information required to construct the reduced density matrices of the CCSD approach: (3)$$\begin{array}{c} \gamma_{1,\;p}^{\mathrm{CCSD}q}=\langle \Phi \left|\left(1+\sum_{\mu }\lambda_{\mu }{\tilde{\tau }}^{\mu }\right){\bar{E_{p}^{q}}}_{0,T}\right|\Phi \rangle , \end{array}$$(4)$$\begin{array}{c} \gamma_{2\;pq}^{\mathrm{CCSD}rs}=\langle \Phi \left|\left(1+\sum_{\mu }\lambda_{\mu }{\tilde{\tau }}^{\mu }\right){\bar{e_{pq}^{rs}}}_{0,T}\right|\Phi \rangle , \end{array}$$
where $e_{rs}^{tu}=E_{t}^{r}E_{u}^{s}-E_{u}^{r}\delta_{t}^{s}$, and the spacial dependency of these quantities is simply recovered using the convolution with the basis-set orbitals in the coordinate representation
$$\begin{array}{c} \gamma_{1}^{\mathrm{CCSD}}\left(r,r^{\prime }\right)=\sum_{pq}\gamma_{1,\;p}^{\mathrm{CCSD}q}\langle r\left|q\right.\rangle \langle p\left|r^{\prime }\right.\rangle ,\\ \gamma_{2}^{\mathrm{CCSD}}\left(r_{1},r_{2},r_{1}^{\prime },r_{2}^{\prime }\right)=\sum_{pqrs}\gamma_{2\;pq}^{\mathrm{CCSD}rs}\langle r_{1}\left|r\right.\rangle \langle r_{2}\left|s\right.\rangle \langle p\left|r_{1}^{\prime }\right.\rangle \langle q\left|r_{2}^{\prime }\right.\rangle . \end{array}$$

The desired densities are obtained as the diagonal part of the corresponding density matrices: $\rho_{1}^{\mathrm{CCSD}}\left(r\right)=\gamma_{1}^{\mathrm{CCSD}}\left(r,r\right)$ and
$$\rho_{2}^{\mathrm{CCSD}}\left(r_{1},r_{2}\right)=\gamma_{2}^{\mathrm{CCSD}}\left(r_{1},r_{2},r_{1},r_{2}\right).$$

Through this construction, the density matrices enter the energy functional as
$$E_{\mathrm{CCSD}}=\int {\left.\left[-\frac{1}{2}\Delta_{r^{\prime }}+V_{en}\left(r^{\prime }\right)\right]\gamma_{1}^{\mathrm{CCSD}}\left(r,r^{\prime }\right)\right|}_{r^{\prime }=r}d^{3}r+\frac{1}{2}\;\int \int \;\frac{\rho_{2}^{\mathrm{CCSD}}\left(r_{1},r_{2}\right)}{r_{12}}d^{3}r_{1}d^{3}r_{2},$$
determining all relevant correlation contributions. To rectify the latter, one should consider the Hartree–Fock counterparts of the expressions above, including the one-electron RDM $\gamma_{1,\;p}^{\mathrm{HF}q}=\langle \Phi \left|E_{p}^{q}\right|\Phi \rangle$, 2-RDM $\gamma_{2,\;pq}^{\mathrm{HF}rs}=\gamma_{1,\;p}^{\mathrm{HF}r}\gamma_{1,\;q}^{\mathrm{HF}s}-\gamma_{1,\;q}^{\mathrm{HF}r}\gamma_{1,\;s}^{\mathrm{HF}s}$, and the related densities $\rho_{1}^{\mathrm{HF}}\left(r\right)$ and $\rho_{2}^{\mathrm{HF}}\left(r_{1},r_{2}\right)=\rho_{1}^{\mathrm{HF}}\left(r_{1}\right)\rho_{1}^{\mathrm{HF}}\left(r_{2}\right)-{\left|\gamma_{1}^{\mathrm{HF}}\left(r_{1},r_{2}\right)\right|}^{2}$. In agreement with the traditional definition, the correlation part of the CCSD energy corresponds to the difference of $\Delta E_{\mathrm{CCSD}}=E_{\mathrm{CCSD}}-E_{\mathrm{HF}}$, which leads to the natural form of the pair density distortions due to the electron correlation $\Delta \rho_{2}^{\mathrm{CCSD}}=\rho_{2}^{\mathrm{CCSD}}-\rho_{2}^{\mathrm{HF}}$. The relevant exchange contribution as the trivial–cumulant part of the pair density, $\rho_{2,X}^{\mathrm{CCSD}}\left(r_{1},r_{2}\right)=\rho_{1}^{\mathrm{CCSD}}\left(r_{1}\right)\rho_{1}^{\mathrm{CCSD}}\left(r_{2}\right)-{\left|\gamma_{1}^{\mathrm{CCSD}}\left(r_{1},r_{2}\right)\right|}^{2}$, can be defined, and the following description for the Coulomb part of the density distortions caused by the correlation can be provided: $\Delta \rho_{2,C}^{\mathrm{CCSD}}=\rho_{2}^{\mathrm{CCSD}}-\rho_{2,X}^{\mathrm{CCSD}}$. Finally, the contributions due to the intermolecular interactions can be related to the supersystem [AB] of two interacting fragments, A and B, which leads to the quantities of the general form
$$\Delta_{\mathrm{int}}\rho_{\left[\mathrm{AB}\right]}=\rho_{\left[\mathrm{AB}\right]}-\rho_{A}-\rho_{B}$$
with $\rho =\rho_{1}^{\mathrm{CCSD}},\rho_{2}^{\mathrm{CCSD}},\Delta \rho_{1}^{\mathrm{CCSD}},\Delta \rho_{2}^{\mathrm{CCSD}},\Delta \rho_{2,C}^{\mathrm{CCSD}},\Delta \rho_{2,X}^{\mathrm{CCSD}}$, respectively.

#### 2.1.2. Cusp Region

Having introduced the relevant densities as the primary factors influencing various correlation effects, we now explore a method to minimize the arbitrariness in selecting the auxiliary basis, aiming for a quicker convergence of intermolecular correlation energies. To achieve this goal, we propose analyzing the dependency of the pair density of electronic coordinates and identifying specific domains where the two-electron density exhibits the most significant behavior contributing to correlation effects. An effective approach involves examining critical and special points within the pair density, considering details such as the function’s shape, angular dependencies, and specific dimensional aspects. This information can then be utilized to finely adjust the parameters of the basis set.

In the case of pair density, the first glance at the problem is greatly facilitated by the cusp-like features at the zeros of the interelectronic distances. The asymptotic behavior of wave functions in the vicinity of these points is controlled via the so-called Kato conditions [56], which are naturally modified for the pair density case [57]. As is illustrated in the helium dimer example by way of an analysis of dispersion function behavior [44,46], compliance with the Kato conditions can be regarded as a good descriptor of the quality correlation treatment, revealing the application of the bond functions [21,30] or explicitly correlated methods [58,59,60] as good practice.

Since the main bulk of the correlation-related specifics of the pair density dependence comes down to the lines of the electron–electron coalescence, a reasonable way to approach the problem of adjusting the bond function basis set starts from an analysis of the diagonal part of the pair density function $\Delta \rho_{2}^{\mathrm{CCSD}}$, on which the most salient minima induced via the correlation should be revealed. This observation was partially confirmed by an inspection of the intermolecular electronic correlation in dispersion-bounded systems such as dimers of noble gases or alkanes [21,30,32,44,45,46]. While the minima points of $\Delta \rho_{2}^{\mathrm{CCSD}}\left(r,r\right)$ would be the first approximation for a place where the bond functions should be centered, the related characteristic exponents and angular momenta of these additional functions can be partially recovered from the relevant angular behavior and radial slopes of this function.

### 2.2. Computational Details

#### 2.2.1. Constructing the Two-Particle Density Matrix

As noted above, a balanced description of the correlation effects using a profound post-HF method is a prerequisite for obtaining chemically relevant results on the intermolecular-correlation-based interactions in weakly bounded systems, including van der Waals complexes, in which the dispersion and induction interactions are pure correlation effects. These originate from the coupling-electron density fluctuations on the fragments due to the intermolecular fluctuation potential.

In this regard, the coupled clusters strategy represents the method of choice if the one-dimensional model space techniques can be engaged, and the CCSD method affords a particularly good cost/performance ratio. In particular, this approach showed good efficiency in the interaction energy calculations for a methane dimer ${\left({\mathrm{CH}}_{4}\right)}_{2}$ and in constructing PES cross sections using the CCSD in combination with the Dunning basis sets cc-pVNZ and aug-cc-pVNZ (N = D, T, Q). This enables us to achieve basis set saturation with cc-pVDZ if the bond functions are used, or with cc-pVTZ, with cc-pVQZ, or without them [23].

For the model system including the methane and Ne dimers, the relaxed density matrices of the CCSD method were constructed in the molecular orbital basis sets using the classical technique outlined in Section 2.1 [48,49,50,51,52]. This construction presumes that the first-order response theory equations [48,49,50,61,62,63,64,65] take the form listed above, which has been solved with respect to the Lagrange multipliers and the wave vector parameters on common grounds.

The reduced two-particle density matrix (2-RDM) in the MO basis was firstly represented in the atomic orbital basis and then recontracted to provide the required coordinate representation as an analytical function of the coordinates of the first and second electrons. Finally, the obtained dependence was corrected on the Hartree–Fock 2-RDM to provide the actual correlation contribution, the diagonal of which had to be examined to locate the most correlated and valuable spatial regions.

#### 2.2.2. Details of the Calculations

The main objects of study were a methane dimer in its D3d geometry [22,23] including an equilibrium one, a neon dimer, and a mixed methane-dimer complex, in which the equilibrium geometries (see Figure 1) were optimized within the same CCSD/basis set combination in which the 2-RDM was considered. Since the pair density calculations were carried out in only one geometry, characterized by quite large interfragment distances, we decided not to resort to the standard counterpoise correction procedure for the target correlation part of the 2-RDM $\Delta \rho_{2}^{\mathrm{CCSD}}$ values. The 2-RDM was constructed in coordinate representation for ${\left({\mathrm{CH}}_{4}\right)}_{2}$ using the CCSD-RDM method with the basis sets cc-pVDZ, cc-pVTZ, and cc-pVDZ+bf[33221], in which the notation bf[33221] means the 3s3p2d2f1g basis on the dummy center recommended in reference [66]. The correlation energies of ${\left({\mathrm{CH}}_{4}\right)}_{2}$ and ${\mathrm{Ne}}_{2}$ were calculated using the same technique with the cc-pVDZ, cc-pVTZ, cc-pVDZ+bf[33221], and cc-pVTZ+bf[33221] basis sets. Moreover, to explore the pair density deficit behavior, due to the correlation in asymmetric cases, the related calculations for the VDZ quality basis set (CH_4_)⋯Ne were carried out for various geometric configurations. Finally, we addressed the influence of the bond functions’ dummy center on the interaction energies in all the mentioned complexes. To this end, the counterpoise-corrected energy dependences [28] on the interfragment distances measured as those between the non-hydrogen atoms were obtained for the cc-pVNZ+bf[33221] (N = D, T, Q) bases with various choices for the dummy center position, and the complete basis set extrapolation was performed [67].

### 2.3. Calculation Results

The expected result of a series of calculations was that, due to symmetry requirements, the main bulk of the pair-electron density deficit would have occurred at the geometric center of the studied dimers, at the midpoint of the C-C or Ne-Ne distances, respectively. Placing an auxiliary set of bond functions at these points, as noted in reference [68], significantly improves the description of the electronic correlation of the compounds under study, which significantly affects the interaction energies of the fragments.

In order to validate these statements, we carried out a series of calculations for the correlation part of the diagonal values of the pair density $\Delta \rho_{2}^{\mathrm{CCSD}}\left(r,r\right)$ and explored regions near the pronounced minimum that arose in the geometric center or the center of mass of the supermolecule. In the case of the methane dimer, these dependences were calculated with the cc-pVDZ and cc-pVTZ basis sets and with sets augmented via bond functions [33221]. The resulting coordinate dependences of pair densities are presented in Figure 2, and the corresponding values at selected points are compiled in Table 1. For the section of pair densities in various bases, we could pinpoint a pronounced minimum at which the bond functions were assumed to be located to reduce the basis set requirements.

In this case, the observed minimum corresponds to the most significant correlation contributions observed in all bases studied. In the interpretation of Coulson [69,70], regarding a fairly complete description of the exchange correlation at the HF level, the pair density minima, as shown in Figure 2, can be rationalized as the point of maximum depth due to the Coulomb hole $\Delta \rho_{2\;C}^{\mathrm{CCSD}}/\rho_{1}^{\mathrm{CCSD}}-\Delta \rho_{1}^{\mathrm{CCSD}}$, the correct description of which is provided via an auxiliary set of functions. Against this background, the explicitly correlated methods can be considered to perform in a similar way, improving the exchange–correlation hole description in two-electron coordinate space, though they are significantly more computationally expensive for large organic systems since these approaches capture the correlation for all coordinate domains, rather than simply where it is most pronounced.

A comparison of the results of this study of methane and neon dimers with the data obtained in another study of a helium dimer [30] confirmed that basis sets containing bond functions allow a more accurate description of the intermolecular electronic correlation. As in the case of the helium dimer, the behavior of the pair density shows a minimum, which corresponds to the middle of the distance between the monomers. In our case, we were able to both construct a graph of the corresponding dependencies and obtain quantitative estimates. In general, for many electron systems consisting of a methane dimer and a helium dimer, it was possible to show that basis sets with bond functions improve the description of intermolecular electron correlation. In addition, the developed tool can be used to study more complex molecules.

Having examined the pair density behavior of the symmetric dimers, in which the dummy center for the additional functions can be naturally chosen for symmetry reasons, we examined the corresponding pair density dependencies for a mixed methane–neon dimer in various geometries using the CCSD-RDM method with the cc-pVDZ basis set. The obtained $\Delta \rho_{2}^{\mathrm{CCSD}}\left(r,r\right)$ sections were shown in Figure 3 for the equilibrium geometry and two relative displacements. As expected, the pair density values turned out to be highly geometry- and system-dependent. In the equilibrium geometry with the C-Ne distance $R_{C-\mathrm{Ne}}$ value of 3 Å, the pair density minimum point was found to be 34% closer to the neon atom, with a relative distance of 0.6 Å, whereas decreasing and increasing the C-Ne distance by 0.4 Å resulted in displacing the minimum by 0.7 and 0.4 Å relative to the Ne position, respectively.

In order to assess the role of the bond functions centering on the quality of the CCSD description for the intermolecular correlation and the extent to which the assumed choice of the pair density minimum is relevant for this, we performed a number of calculations for the counterpoise-corrected interaction energy depends on the interfragment distance of the methane and neon dimers, as well as the methane–neon complex, according to the procedure described in Section 2.2.2. The CCSD calculations were carried out with the aug-cc-pVNZ (N = D, T, Q) bases, followed by the CBS extrapolation with the bond functions located at various selected positions, including the pair density minimum point recovered using the aug-cc-pVDZ basis. The benchmark data were obtained in a similar way using the aug-cc-pVNZ (N = D, T, Q, 5, 6) basis set series without the bond functions with subsequent CBS extrapolation.

Special attention was paid to the potential energy curves of the asymmetric (CH_4_)⋯Ne complex for which the bond functions were placed in the geometric center of the C-Ne bond, shifted relative to the center of the carbon atom and in the perpendicular direction in 1 Å and at the pair density minimum point. The results shown in Figure 4 confirm that an inconsistent expansion of the small basis sets in general deteriorated the interaction energy data, improving the description only for a special choice of the auxiliary function parameters that, for the bond functions, were determined by the pair density behavior. In particular, due to the balanced BSSE cancellation in the counterpoise procedure for the bond functions centered on the pair density minimum, the best performance in terms of the basis set saturation was already observed for the cc-pVTZ level (see Figure 4). In turn, this contrasts with the conclusion drawn for noble gas diatomics in [34], in which dummy center displacements did not significantly affect the result for converged energy values. At the minimum point, the energy was equal to 0.13 kcal/mol, and the distance between the carbon and neon atoms was 3.4 Å. The differences between the curves constructed with the cc-pVTZ, cc-pVQZ, and CBS basis sets were quite small, and they amounted to up to 1% in the equilibrium distance. The same observations are valid for comparisons with the benchmark data obtained without the auxiliary set, which showed the almost instantaneous convergence of the results mentioned for the minimum pair density placement. The opposite observation could be made for the other choices of the dummy center position, shown in Figure 4. The inconsistency in the fragments and the complex description resulted in quite irregular behavior for the interaction energies when moving from the cc-pVDZ to the cc-pVQZ bases. Moreover, although the comparison of the curves for the cc-PVTZ and cc-pVQZ bases assumed that basis set saturation had occurred at the cc-pVTZ level, the unbalanced description of the monomers relative to the full complex led to severe artifacts when extrapolating to the CBS limit. In particular, when the distance was reduced to 3.8 Å, a sharp change in the interaction energy dependence was observed for cc-pVDZ; the values of the interaction energy took excessive values, especially at the minimum distance of about 3.4 Å. Hence, the correspondence between the basis size and the energy value was violated, and the energies underwent abrupt changes in the cc-pVTZ, cc-pVQZ, and cc-pVDZ series.

Finally, we addressed the unified treatment of the relative errors and convergence rate dependencies at the dummy center position for the (CH4)⋯Ne interaction energies. To this end, we investigated the average values of $\epsilon_{x}=\left|E_{x}-E_{\mathrm{CBS}}\right|/\left|E_{\mathrm{CBS}}\right|$ and $\varepsilon_{x}=|\epsilon_{x+1}-\epsilon_{x}\left|/\right|\epsilon_{x-1}-\epsilon_{x}|$ descriptors with *x* = 3, 4, 5, and “*∞*” in the region $R_{{\mathrm{CH}}_{4}-\mathrm{Ne}}=2.8,3.0,\cdots ,10$ Å near the equilibrium geometry and at the intermediate intermonomer distances, recovered for the aug-cc-pVNZ (N = D, T, Q) bases, and the related CBS extrapolations. We explored a wide range of dummy center displacements from 0.2 to 1 Å relative to the center of the intermonomer distance and at the pair density minimum obtained without the bases’ augmentation. As a result, it was found that adding the bond functions at the minimum position accelerated convergence in the sense of the specified descriptors by about 20%, reducing the average errors relative to the CBS results for a given shift by 11% and up to 24% relative to CBS at the minimum pair density point. A similar treatment was applied to the symmetric cases ${\left({\mathrm{CH}}_{4}\right)}_{2}$ and ${\mathrm{Ne}}_{2}$, which led to analogous results, generally facilitating the convergence by an average of 34% and 26% for the neon and methane dimers, correspondingly. Thus, the example of the methane–neon system shows that an inadequate choice of bond function parameters leads to an unbalanced description of the intermolecular correlation effects. In this case, the auxiliary functions resulted in the deterioration of the cc-pVDZ basis results, which were partially remedied on the cc-pVQZ level. The equilibrium interaction energy value of 0.13 kcal/mol reproduced for the pair density minimum results revealed errors of more than 5% for the considered alternative choices of the dummy center.

Similar conclusions can be drawn for the symmetric methane and neon dimers, for which the potential energies are shown in Figure 5, where the pair density minimum choice for the dummy center already led to basis saturation for cc-pVDZ and cc-pVQZ sets, resulting in a binding energy of 0.39 kcal/mol for the methane dimer and 0.063 kcal/mol for the neon dimer, respectively. Displacing the center along the equatorial plane required much larger bases for basis set saturation; specifically, the binding energy was underestimated at the cc-pVTZ and cc-pV5Z levels, resulting in values of 0.36 kcal/mol and 0.057 kcal/mol for the methane and neon dimers calculated using the CCSD method, respectively, which corresponded to the relative error exceeding 5%. A further deterioration of the results was observed with the shift towards a monomer position. In this case, the imbalanced BSSE cancellation completely disrupted the convergence, making the binding energy values for the methane and neon dimers equal to 2 kcal/mol and 7 kcal/mol, respectively, for the cc-pVTZ basis set, and even worse results of 14 and 30 kcal/mol, correspondingly, were observed for the cc-pVDZ level.

## 3. Methods

To perform CCSD 2-RDM calculations, we employed the algorithm proposed by Gauss et al. [48] with the following contraction in order to obtain the relevant coordinate representation; for this, the C++ program package was implemented. In the main pipeline, the $T_{1}$ and $T_{2}$ amplitudes were recovered from the iterative solution of the CCSD equations, alternating with solving linear equations on Λ amplitudes. The canonical molecular orbitals, cluster amplitudes, and all the necessary molecular integrals were ascertained via calculations through the MOLPRO software package (version 2010.1) [71].

The system of equations for the Λ amplitudes (Λ–equations) was solved using the GMRES method with a precondition [72,73] for which the DIIS technique was used to accelerate the convergence [74,75]. The Krylov space method GMRES was preferred over the usual Jacobi method since it revealed itself as a much more robust technique, particularly in cases of non-diagonally dominant equation systems for which the Jacobi approach was designed [72]. The main disadvantage of this method, slowing its convergence, was eliminated by using the Jacobi precondition [76] and by addressing the DIIS approach for each outer iteration of GMRES.

The matrix of molecular integrals was stored symmetrically since it requires a large amount of computer memory and is symmetrical with respect to electronic permutations. The same problems of allocating and storing for large sparse matrices arose for the cluster amplitudes for which the effective caching method was applied.

## 4. Conclusions

The results outlined in the previous section illustrate that the correct choice of the auxiliary function parameters, particularly, the dummy center of the bond functions set, is mandatory in order to obtain converged results for the weakly bonded systems, in which the direct increasing basis set turned out to be prohibitively expensive. The reasonable option for tuning these parameters on the bond functions example was proposed based on the pair density features that locate the spatial domain responsible for the major bulk of the dispersion energy or, generally, the correlation-based properties.

The calculations in the present work carried out for the molecules Ne_2_, ${\left({\mathrm{CH}}_{4}\right)}_{2}$, and ${\mathrm{CH}}_{4}\cdots \mathrm{Ne}$ were intended to represent the diagonal pair density features as a good descriptor of the local correlation effect analysis in the cases in which geminal-dispersion-function-based considerations lacked universality. The well-known fact of applying the bond functions method to improve the intermolecular correlation description was verified for the model systems concerned. Moreover, it was found that the bond functions at the pair density minima greatly facilitated the basis set convergence, at least for the interaction energy results. This observation can be attributed to the local Coulomb hole dependence revealing maxima in the bond centers for the equilibrium geometries [69,70].

The presented approach to choosing the dummy center, as well as further assumptions on the close relationship between the two-particle density features and the criteria for replenishing the basis set, can allow us to establish a well-defined procedure for tuning the auxiliary basis parameters that is devoid of classical arbitrariness problems and commonly addressed using different heuristics [21,35].

## Figures and Tables

**Figure 1 ijms-25-01472-f001:**
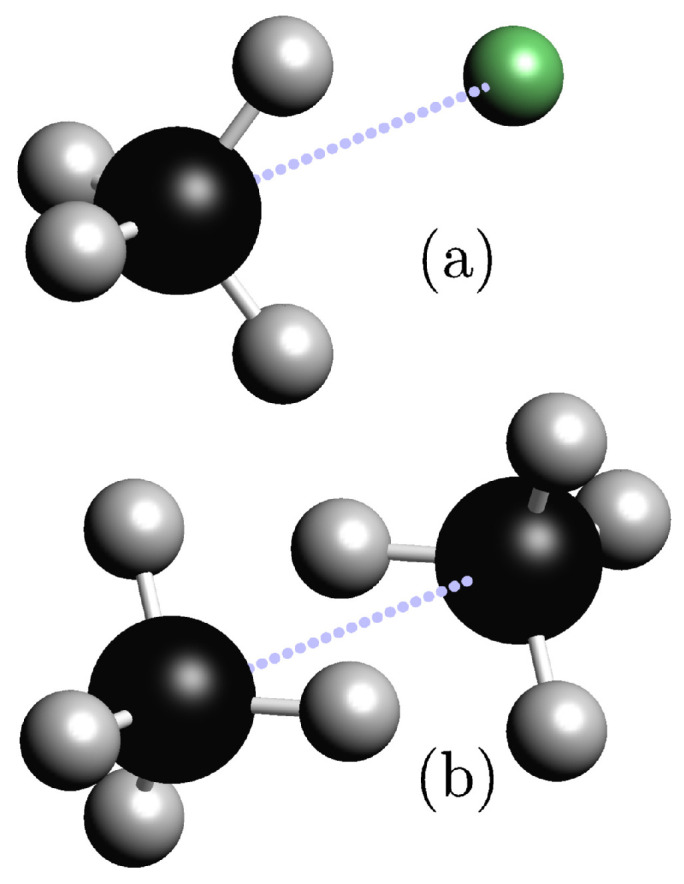
Schematic representation of the methane–neon complex (**a**) and methane dimer (**b**) equilibrium geometries.

**Figure 2 ijms-25-01472-f002:**
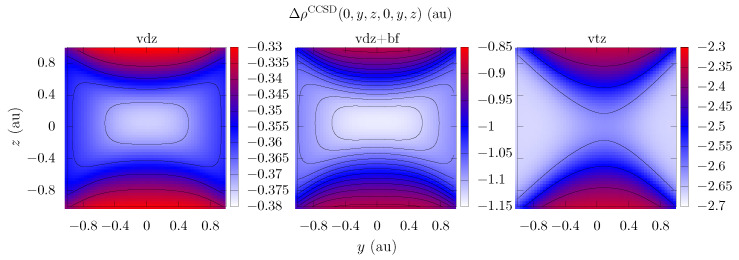
The sections of the coordinate diagonal pair density dependence for the methane dimer along the yz plane $\Delta \rho_{2}^{\mathrm{CCSD}}\left(0,y,z,0,y,z\right)$; the captions vdz, vdz-bf, and vtz correspond to the cc-pVDZ, cc-pVDZ+bf[33221], and cc-pVTZ basis sets, respectively.

**Figure 3 ijms-25-01472-f003:**
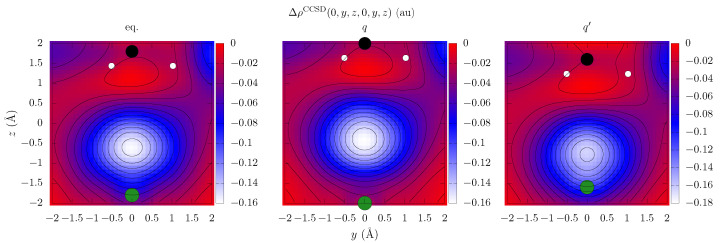
The section of the diagonal pair density coordinate dependence $\Delta \rho_{2}^{\mathrm{CCSD}}\left(0,y,z,0,y,z\right)$ in the yz plane for the (CH_4_)⋯Ne complex obtained from the CCSD-RDM/cc-pVDZ calculations. The captions eq, q, and q’ correspond to the equilibrium geometry and the geometries with the Ne-C distance being increased and decreased by 0.4 Å, respectively. The green, black, and white circles depict the projection of the neon, carbon, and hydrogen atoms, correspondingly.

**Figure 4 ijms-25-01472-f004:**
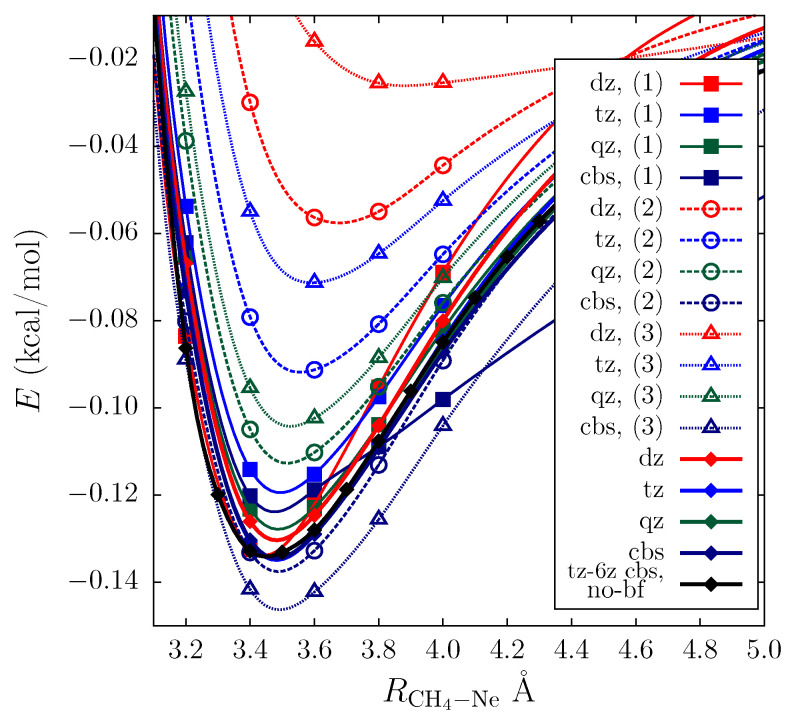
Interaction energy dependences of the (CH_4_)⋯Ne complex obtained in the CCSD calculation for the cc-pVNZ+bf (N = D, T, Q) bases and the corresponding CBS results. The CBS extrapolation data for a series of TZ-6Z basis sets without the bond functions are depicted using a solid black line with diamonds. The best results observed when placing the bond functions at the pair density minimum are shown using solid lines with diamonds. The curves obtained with bond functions positioned at the geometric center of the C-Ne bond or shifted relative to one by 1 Å along the equatorial x direction or towards the carbon atom are indexed as (1), (2), and (3), respectively.

**Figure 5 ijms-25-01472-f005:**
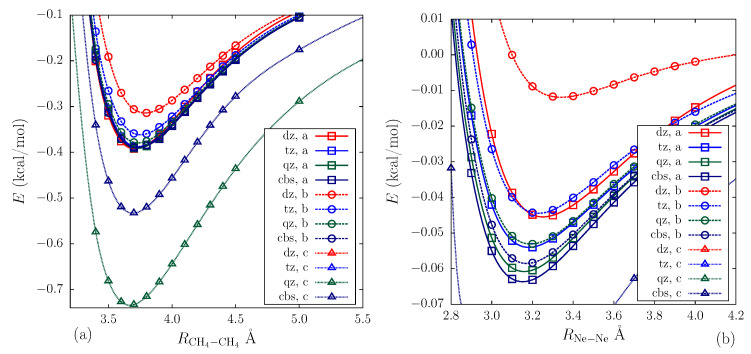
Interaction energy dependences of the (CH_4_)_2_ and Ne_2_ molecules on the left panel (**a**) and the right panel (**b**), respectively, obtained via the CCSD calculation for the cc-pVNZ+bf (N = D, T, Q) bases and the corresponding CBS results. The curves obtained with bond functions centered on the geometric center or shifted relative to one by 1 Å along the equatorial x direction or towards one of the fragments are specified with the a, b, and c labels on the legend, correspondingly.

**Table 1 ijms-25-01472-t001:** The diagonal values of the pair density matrix of ${\left({\mathrm{CH}}_{4}\right)}_{2}$ corrected on the Fock contribution $\Delta \rho_{2}^{\mathrm{CCSD}}\left(r,r\right)=\rho_{2}^{\mathrm{CCSD}}\left(r,r\right)-\rho_{2}^{\mathrm{HF}}\left(r,r\right)$ in various points with respect to the equilibrium geometry optimized in CCSD, in which the dimer was oriented on the ζ molecular axis, for the different basis sets used.

Electron Coordinates r=(x,y,z) (a_0_)			cc-pVDZ	cc-pVTZ	cc-pVDZ +bf[33221]
* **x** *	* **y** *	* **z** *	$\Delta \rho_{2}^{\mathrm{CCSD}}\left(r,r\right)$		
0	0	0	−0.4143	−2.71	−1.26
0	0	0.9	−0.321	−2.66	−0.808
0	0.9	0	−0.355	−2.33	−1.06

## Data Availability

The CCSD-RDM-CR software package (version I) was developed for constructing CCSD-RDM in coordinate representation. The public repository can be found at https://gitlab.com/bogdanr1/ccsd-rdm-cr.

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
