# Peer review of "The Role of Bond Functions in Describing Intermolecular Electron Correlation for Van der Waals Dimers: A Study of (CH_4_)_2_ and Ne_2"

_ijms, 2024, doi:10.3390/ijms25031472_

Round 1

Reviewer 1 Report

Comments and Suggestions for Authors

The authors present a study on the role played by midbond functions in the description of intermolecular correlation energy in methane and Ne dimers.

Even though the subject could be of interest I find several flaws in the manuscript as listed below:

1)    I find the manuscript ill-structured. The introduction and Theory sections are in my opinion too large, describing for the most of them already known procedures and theoretical methods. These sections could be greatly reduced by using appropriate references.

2)    In contrast to my previous point, the results and discussion section is extremely short. Just two cases are presented for a given geometry, showing that, as already known, midbond functions improve the description of intermolecular interaction by helping the saturation of the one-electron basis set.

3)    Finally, the language should be improved, and the references must be checked for correction (for instance ref. 22 has nothing to do with methane dimer).

In summary, I do not find that the manuscript provides a significant contribution to the subject, since it is limited to a very small sample of dimers/geometries, and just follow the behavior already described in literature. Thus, the manuscript does not provide new insights to merit its publication in IJMS.

Comments on the Quality of English Language

A general language revision is needed. many sentences and paragraphs are difficult to follow.

Author Response

We appreciate the first reviewer's insightful comments and largely concur with their overall assessment. However, we would like to offer some clarifications on specific points to enhance the clarity and coherence of our work.

Regarding the text structure, it should be pointed out that the theoretical section deliberately focuses on providing a broad understanding of acceptable options for approximating the relevant correlation characteristics, rather than solely detailing the specific approach employed. Additionally, the variations in the actual form of the traditional response theory working equations across different studies necessitates explicitly specifying the equation form addressed in the current research to facilitate transparency and reproducibility.

The introduction's primary purpose is to clearly establish the focus of the study, which lies on validating the relationship between the geometric characteristics of the hole function and the tuning of the auxiliary basis parameters, especially, the dummy atomic center coordinates. This goes beyond merely demonstrating a correspondence between basis set extension and improved electronic correlation description.

Our study provides compelling evidence that incorporating a floating basis centered at the pair density minima significantly enhances convergence for a specific variational strategy, particularly with respect to the size of the basis set. This finding represents a valuable contribution since it establishes the way in which the auxiliary functions can be designed and added avoiding the generic arbitrariness of these procedures as well as the heuristics to cope with that arbitrariness. In particular the second manuscript version presented contains the analysis of the hole behavior--the bond functions parameters for the asymmetric bimolecular dispersionally bounded complexes, which exemplifies the general correspondences highlighted in the work results on the optimal way for the bound functions placing. The supplemented material is highlighted in red and presented in the file Bogdan_Rutskoy_english-74878-edited.pdf.

Finally, with respect to the work language and the references, we acknowledged the reviewer's feedback and fixed the references problem, as well as addressed the language issue during the meticulously reviewing and revising our work.

Reviewer 2 Report

Comments and Suggestions for Authors

See attached

Comments on the Quality of English Language

Sentences are usually too large and hard to follow. Sometimes this produces grammatical inconsistencies

Author Response

We acknowledge the reviewer's point that chemical problems often involve larger systems or specific states of small molecules under external conditions. However, it is important to emphasize that our work does not focus on directly calculating chemically or biologically relevant systems. Instead, it aims to demonstrate a novel basis set calibration method, using small weakly bound molecules as model systems. This method allows us to efficiently capture correlation effects observed in molecular systems of any size. Specifically, the established relationship facilitates accelerated calculations for large molecular fragments. These fragments are crucial for parameterizing force fields, which are then used to analyze the properties of systems at chemically and biologically relevant scales.

We as well as addressed the language issue during the meticulously reviewing and revising our work. 

Reviewer 3 Report

Comments and Suggestions for Authors

As can be seen by the above answers to the detailed questions presented for the review, the paper is well written and the results are well described. I think the paper should be published with the one required addition listed below.

The authors appropriately consider the basis set superposition error (BSSE) for the overall calculation. A 2000 JPCA paper  calculates and discusses the BSSE for a number of different calculational algorithms (including ones employed in the present submission). This paper should be referenced and even discussed in the Introduction, as it covers some of the ideas presented in this submission. 

Other than this one suggestion, the paper should be published in the IJMS. 

Author Response

We deeply appreciate the reviewer's insightful comments and acknowledge the importance of addressing the BSSE issue. As the reviewer rightly pointed out, BSSE presents a significant challenge to accurate ab initio studies of weakly bound molecular systems.

In the second edition of the manuscript we added a link to a Dunning et al., 2000, research, on the molecular binding energies calculations including the approaches to tackle the BSSE problems for weakly bound molecular systems.

Round 2

Reviewer 1 Report

Comments and Suggestions for Authors

The authors have partially addressed some of my previous concerns, but I am still not satisfied with the present manuscript that, in my opinion, it still shows several important flaws.

1)    References. The authors have partially corrected some errors in the references, but there are still some improvements to be made. The full set of references should be thoroughly revised (maybe there is no need of so many general references in the Introduction and just some key general ones should be enough)

-       For instance, in page 1 the authors indicate that “For this case, a basis set saturation  (with up to 1% accuracy) has been achieved for the CCSD(T) method in the aug-cc-pVTZ  basis set, yielding an interaction energy of 0.519 kcal/mol [22].” However, there are no CCSD(T) calculations in ref. 22, so the authors should cite the proper references supporting this sentence (maybe Tsuzuki et al. JCP 124 (114304) 2006 ?)

-       Also, I find the manuscript lacking of references and discussion of significant previous work on midbond functions that is very important for the present issue.  See, for instance, Matveeva et al. J. Comput. Chem.  Doi: 10.1002/jcc.26777 and references therein,  where the effect of midbond functions is studied at the MP2 and CCSD(T) levels. Also, Tao J. Chem. Phys. 1993, 98, 3049 and Mol. Phys. 1994, 81, 507 show that “interaction energy is highly insensitive to the displacement of midbond functions and thus the geometric midpoint of the van der Waals bond is a convenient and suitable choice for the position of midbond functions”; Shaw and Hill Mol. Phys. 2018, 116, 1460 also studied the effect of the location of the midbond functions on the interaction energy. I think that these studies are of central importance to the matter at hand and should be properly acknowledged and discussed.

-       Finally, when talking about supermolecular and perturbational (SAPT) approaches to the interaction energy, proper references should be included.

2)    Regarding the results, the authors indicate several times that the use of midbond functions improve the description of intermolecular correlation, but this is already known from previous work. Also, the authors claim to have demonstrated that the optimal position of the midbond functions is on the minimum of the two-particle electron density. I can see that there is a minimum on the two-particle electron density, matching the geometric center in homodimers and displaced in the newly included heterodimer CH4-Ne. However, I do not see whether the authors have checked different positions of the midbond functions in order to conclude that the optimal position is on the minimum of the two-particle electron density (in fact, previous studies indicate that the position of the midbond functions is almost irrelevant). So, I see that the authors demonstrate that there is a minimum in the 2-RDM located in the line connecting the monomers, but I do not feel that they have demonstrated that it is the optimal location for midbond functions.

3)    The authors include Tables 2 and 3, but nothing is said about them.

4)    I still believe that the section describing theory (mostly CC) is too large.

Comments on the Quality of English Language

Just some typos; for instance, "metal" instead of methane.

Author Response

We express our gratitude to the reviewers for their detailed interest in our work. For the convenience of the next review, we have marked in red color new fragments of the text of the manuscript.

We have made every effort to address the identified shortcomings. The described methane dimer calculation conducted in the CCSD(T) / aug-cc-pvtz method is related to the work by Rutskoy, B.V.; Bezrukov, D.S. ("Ab Initio Description of the Structure and Interaction Energy of Perhalomethane Dimers. Russian Journal of Physical Chemistry A 2019, 93, 1519–1524. https://doi.org/10.1134/s0036024419080259") [24] and ("Journal of Molecular Structure: THEOCHEM 897 (2009) 90–94) https://doi.org/10.1016/j.theochem.2008.11.026" [23].

We have analyzed and considered the articles mentioned in the introduction and discussion of results: R. Matveeva et al., J. Comp. Chem. 2021, 43; F.M. Tao. J. Chem. Phys. 1993, 98, 3049; Shaw and Hill. Mol. Phys. 2018, 116, 1460.

In particular, the calculation of asymmetric dimers in the work (3) includes a comparison of cases using basis sets containing bond functions centered on the bond midpoint and without them. Although the bond function technique represents certain advantages, there is a high level of arbitrariness in the choice of the parameterization including the dummy center 66 position as well as the auxiliary set size. Despite the fact that for some noble gas dimers the bond functions centering issue was found to be of minor importance, the situation 68 turns out to be more complex in the general case. 

The optimization of bond function localization at the MP2 level in the work (4) shows that their center shifts towards the heavier monomer. regular analysis of the auxiliary set parameterization (4) should take into account the effects of successive basis extension, 73 optimizing the integral correlation characteristics under question with respect to the auxiliary function positions and exponents, throughout the entire geometric parameters domain of interest, which becomes too computationally expensive. That is why the main efforts focused on optimizing the set of bond functions and Gaussian exponents. This makes the question of finding the optimal bond function localization relevant. The other two works address general issues related to the importance of bond functions. References to them are provided.
The text includes a reference related to the connection between intermolecular interactions and the accuracy of potentials computed by many-body SAPT, citing the work "Chem. Rev. 1994, 94, 1 887-1 930."
To examine the sensitivity of electronic correlation to bond function localization, a series of PES cross-sections for neon-methane dimers and the methane-neon system was constructed. The discussion of the results shows that improvement in description is achieved when bond functions are centered at the point of minimum electronic density.

Regarding the remark about the tables - unfortunately, in the previous version of the text, tables with intermediate results were added, which do not carry much meaning. We have removed this from the text of the manuscript.
We also tried to shorten the theoretical part of the section describing theory, replacing some of the calculations with references to the literature, where they can be read in more depth if desired.

Round 3

Reviewer 1 Report

Comments and Suggestions for Authors

The authors have considered most of my previous suggestions, finally providing results about why locating midbond functions on the minimum of the two-particle electron density, though I would like a more direct comparison. For instance, in figure 4 I believe it would be more appropiate if, for instance, the CBS results obtained at different locations were in the same plot. Also, the authors should explain more clearly how they decide that the CBS values with midbond functions located in the minimum are better than those obtained with midbond functions located at other positions.

As a general comment, I think that the authors did not stress the focus of their results in previous versions of the manuscript; that is, that locating midbond functions in the minimum of the two-electron density is theoretically sound and beneficial for the results obtained as compared with other possible locations. I think that in the present form, this is finally more clearly stated.

Author Response

We are very grateful to you for the critical comments you provided.

Since our work was primarily focused on proposing a rather simple modification to the general methodology to handle with the bond functions, besides the other text issues, we unintentionally made a number of oversights and missed several important details. In the current version of the article, we have included the results of reference calculations for an asymmetric dimer without the basis augmentation, confirming general conclusions which are well-known for symmetric dimers. In addition, we have considered a more general setting, relying on the interaction energy as the main descriptor. Specifically, for the interaction energy curves and various displacements of dummy center, we analyzed the average errors in energy estimation and extrapolation as well as the convergence rate. Despite the fact that all criteria for such assessment assume some arbitrariness in choosing the actual descriptors and the averaging technique, they still allow for a qualitative analysis, the results of which are briefly presented in the text. The corresponding insertions and modifications to the text are highlighted in blue.